# Cell-Mediated Therapies to Facilitate Operational Tolerance in Liver Transplantation

**DOI:** 10.3390/ijms22084016

**Published:** 2021-04-13

**Authors:** Samia D. Ellias, Ellen L. Larson, Timucin Taner, Scott L. Nyberg

**Affiliations:** 1Division of Transplant Surgery, Department of Surgery, Mayo Clinic, Rochester, MN 55905, USA; ellias.samia@mayo.edu (S.D.E.); larson.ellen1@mayo.edu (E.L.L.); taner.timucin@mayo.edu (T.T.); 2Department of Immunology, Mayo Clinic, Rochester, MN 55905, USA

**Keywords:** liver transplantation, transplant immunology, operational tolerance, regulatory T cells, mesenchymal cells, dendritic cells

## Abstract

Cell therapies using immune cells or non-parenchymal cells of the liver have emerged as potential treatments to facilitate immunosuppression withdrawal and to induce operational tolerance in liver transplant (LT) recipients. Recent pre-clinical and clinical trials of cellular therapies including regulatory T cells, regulatory dendritic cells, and mesenchymal cells have shown promising results. Here we briefly summarize current concepts of cellular therapy for induction of operational tolerance in LT recipients.

## 1. Introduction

Transplantation of organs, including the liver, across the HLA barrier induces strong alloimmune responses in recipients. Both cellular and humoral alloresponses contribute to rejection. Current immunosuppressive (IS) therapies including calcineurin inhibitors (cyclosporin and tacrolimus) and corticosteroids target recipient T cell activation, expansion, and cytotoxicity effectively, and reduce acute rejections to less than 15% of transplants but their long-term use is associated with increased risk of infections and malignancies [1,2]. While our understanding of the role of antidonor antibodies in acute and chronic rejections has improved significantly over the past decade, existing therapies such as anti-CD20 or complement pathway monoclonal antibodies are only partially effective and do not prevent the development of chronic antibody-mediated injury [3].

More than half a century ago, Billingham and Medawar described the phenomenon of acquired immunologic tolerance to transplant antigens by successfully grafting the skin of a calf onto its fraternal twin [4]. Induction of immune tolerance decreases the risk of graft rejection after solid organ transplantation and thus reduces the need for immunosuppression and improves the survival of transplanted organs. Billingham’s work was followed by the first successful kidney transplant in 1954, and so launched the worldwide search for methods to induce immune tolerance and to hold graft rejection at bay [5]. Transplant tolerance represents the holy grail for transplant immunology: a state where the allograft is accepted by the recipient in the absence of IS treatment. There are multiple types of tolerance including full immunological tolerance, operational tolerance (OT), or IS minimization, sometimes referred to as “prope tolerance” [6]. In the practical setting, research is focused on the induction of OT, which is defined as stable graft function in the absence of IS for more than one year without any features of chronic rejection [7]. Efforts to develop OT have capitalized on how the immune system actively regulates itself through regulatory T cells, B cells, and innate components. Studied methods for developing OT include hematopoietic stem cell transplantation that re-educates the immune system and targeted stimulation by transfer of immune regulatory cells [8]. Antigen-presenting cells (APCs), Kupffer cells (KCs), non-immune cells such as mesenchymal cells (MSCs), hepatic stellate cells (HSCs), and liver sinusoidal endothelial cells (LSECs) have regulatory properties as well [9]. The use of cells as therapeutics, particularly regulatory T cells (Tregs), regulatory dendritic cells (DCregs), and MSCs have attracted enthusiasm as alternative OT induction strategies in transplants. The current review will address the role these cells play in liver graft tolerance and summarize the current status of clinical therapies utilizing them.

## 2. Historic Overview

Liver transplantation is presently the only proven treatment for end-stage liver disease. Liver transplant was first performed clinically by Starzl in 1963 [10]. In 1969, Calne recognized that pigs could survive without IS for long periods after liver transplantation, and that the liver graft could provide a protective benefit against rejection to other solid organs transplanted from the same donor [11]. Further study of the liver’s protective effect on kidney transplants was undertaken in human simultaneous liver-kidney transplant recipients. These studies showed reduced chronic cellular and antibody-mediated alloimmune injury and lower overall frequency of circulating CD8+, CD4+, and effector memory T cells on flow cytometry, as well as upregulated genes and gene sets associated with tissue integrity and metabolism in kidneys transplanted simultaneously with a liver, compared to solitary kidney recipients [12,13,14].

These and similar observations opened the gateway to IS weaning in LT which could address the long-term toxicities and cost of therapy. A meta-analysis summarized the accumulated clinical experience with IS-weaning protocols in LT from 1993–2006 and the available data indicated that elective IS withdrawal is possible in 19.4% of recipients [15]. However, selection bias likely contributed to this rate, as some of these studies chose participants who were more likely to be successfully weaned off IS, and no intention-to-treat weaning trials were included. Hence it is difficult to accurately estimate the actual rate of OT. A study conducted by Feng et al. in 2012 showed 60% OT in a pediatric population. However, this study required a longer time from transplant (four years vs two years) and excluded those with autoimmune or viral diseases [16]. Despite these results, the long duration to wean recipients’ off IS and the lack of generalization, inducing tolerance in LTs has gained interest and multiple OT induction trials such as Treg cell therapy are currently conducting clinical trials to induce OT in LTs [17]. NCT03577431 and NCT03654040 are Phase I/II studies currently recruiting participants who will receive Tregs; those that successfully discontinue IS for 52 weeks will be defined as operationally tolerant.

Over the past three decades, researchers have been studying ways to induce tolerance to solid organ transplants using hematopoietic stem cells. In theory, donor bone marrow infusion should produce donor pluripotent hematopoietic stem cells, inducing donor-specific hyporeactivity [18]. Rao et al. transplanted LT recipients with donor bone marrow hematopoietic stem cells harvested from vertebral bodies; HSC recipients showed 62% donor-specific hyporeactivity as compared to 21% seen in LTs only [19]. Donckier et al. succeeded in early IS withdrawal in two out of three LT patients infused with donor bone marrow hematopoietic stem cells under myeloablative conditioning [20]. Nevertheless, myeloablative conditions are necessary for the engraftment of donor bone marrow and carry a risk of toxicity to the recipient. Tryphonopoulos et al. reported that donor bone marrow hematopoietic stem cell infusion under non-myeloablative conditions in LT recipients showed no increase in the likelihood of IS weaning [21]. Simultaneous donor bone marrow hematopoietic stem cell infusion with LTs increases the risk of graft versus host disease (GVHD) when compared to LTs alone; two out of 29 patients from Rao’s study group developed GVHD [19].

Immune cell therapies have been shown to induce stable tolerance in clinical liver transplantation and several transplant centers are conducting tolerance-inducing clinical trials in LT recipients [17]. They are considered better alternatives as they do not require the rigorous myeloablative conditioning required in hematopoietic stem cell transplantation [22]. In addition, GVHD and over-immunosuppression sequelae such as opportunistic infections are less likely to develop even when cellular therapies are expanded under polyclonal conditions [23].

## 3. Liver’s Unique Tolerogenicity and the Role of Liver-Resident Cells

It is well recognized that the liver is a uniquely immunologically privileged organ [11]. LTs require the least amount of induction and maintenance IS and sustain the lowest incidence of chronic immune-mediated injury [11]. Post-transplant, the liver’s resistance to antibody-mediated rejection could be due to its dual blood supply from the high-pressure systemic and low-pressure portal circulation which meet at the fenestrated sinusoids and facilitate interaction between antigens, T cells, and other resident immune cells. Taner et al. hypothesized that this protects the liver allograft from rejection as the standard capillary microvasculature and single afferent blood supply in other organs get occluded easily by complement-activated immune complex resulting in ischemia [24,25]. Secondly, innate immune cells in the liver express undetectable levels of major histocompatibility complex (MHC) antigens and costimulatory molecules which make it difficult for them to induce an immune response [26]. Unlike other organs, if T cell-mediated rejection episodes occur in compliant liver recipients, they occur more frequently within six weeks post-transplant and do not appear to have a long-term effect on the survival of the allograft [24,25]

Thirdly, there is a large population of migratory immune cells in liver allografts compared to other solid organs that can further explain its privileged status. In fact, the liver can be described as a lymphoid organ [27]. Immune cells of lymphoid and myeloid lineage line the sinusoids in the liver and are involved in tissue homeostasis and immunoregulatory activities. An average adult liver of 1.5 kg is likely to contain 10^10^ lymphocytes [28]. These cells include T cells, natural killer cells (NK cells), and natural killer T cells (NK T cells) (Table 1). Most of the liver is composed of hepatocytes and occupies 78% of the parenchymal volume, while the non-parenchymal cells (NPCs) account for 6% of the volume. The extracellular space, including sinusoidal lumen, space of Disse, and biliary canaliculi together occupy the remaining 16% [29]. NPCs of the liver include the hepatic stellate cells (HSCs), Kupffer cells (KCs), liver sinusoidal endothelial cells (LSECs), dendritic cells (DCs), and intrahepatic lymphocytes [28,29].

Finally, the liver is constantly exposed to gut-derived pathogens and antigen metabolic byproducts which is evident from high levels of lipopolysaccharides (LPS) in the portal system. DCs, KCs, and LSECs expressing toll-like receptors (TLR) recognize pathogen-associated molecular patterns (PAMPs) as part of their innate defense from foreign pathogens. In response to TLR stimulation due to constant exposure to low levels of LPS, these cells release anti-inflammatory cytokines such as IL-10. This phenomenon is known as LPS tolerance and TLR signaling plays a critical role in maintaining the balance between immune activation and tolerance [32]. This concept of tolerance is maintained by sinusoidal APCs—LSECS, KCs, and DC—and implies that the intravascular compartment of the liver is an anatomical structure that supports the deliberate induction of tolerance [33]. This could further explain the discrepancy between the inherent tolerogenicity of the whole liver versus isolated hepatocytes and the contribution of sinusoidal APCs to liver tolerance [34]. These mechanisms of NPC and immune cell roles in liver tolerance are under investigation; [8] some of their mechanisms are presented below.

## 4. Liver MSCs

Liver MSCs have been isolated from biopsies of livers prepared for transplant and have a higher expression of PD-L1 and PTGS1 which are known inhibitors of T cell proliferation compared to bone marrow and adipose-derived MSCs [35]. They are more potent inhibitors of alloreactive T cell proliferation, NK cell cytotoxicity, and IFNγ production compared to MSCs from other sources [35]. Liver MSCs have increased transcriptional activity of hepatocyte growth factor which is known to have anti-T cell proliferative properties [36]. Thus, the limited studies to date suggest that liver MSCs are uniquely equipped immunomodulatory cells and have a potential cell therapeutic application for tolerance induction in transplant settings.

## 5. Liver DCs

Multiple studies to date demonstrate that liver DCs possess superior tolerogenic properties compared to their counterparts isolated from other tissues. For example, freshly isolated liver DCs induce higher numbers of Tregs than their blood-derived DCs [37]. In addition, these cells generate less proinflammatory cytokines such as IL-1 and TNF-α and more anti-inflammatory cytokines including IL-10 [37]. Overall, isolated human liver DCs promote T cell hyporesponsiveness after Ag-specific stimulation which supports the notion that liver DCs are critical regulators of liver tolerance after transplantation [37]. Liver-derived donor DCs in a rat model prolonged pancreatic islet allograft survival [38]. Liver DCs on stimulation with GM-CSF exhibited low levels of MHC II and failed to stimulate allogenic T cells in mixed lymphocytic reactions (MLRs) [38]. This study suggests that liver DCs may contribute to tolerance in liver allografts as well as protect other allografts of the same donor from rejection [38].

## 6. HSCs

HSCs are pericytes in the space of Disse, between the sinusoids and basal surface of the hepatocytes. They play a role in maintaining the extracellular matrix, sinusoidal blood flow, storage of vitamin A, and function as APCs [39]. On stimulation with IFNγ, they express CD1d, CD86, and MHC II. Multiple mechanisms are involved in the immune tolerance induced by HSCs, namely the expansion of myeloid-derived suppressor cells (MDSCs). MDSCs and all-trans retinoic acid in the HSCs express high levels of arginase 1 and inducible nitric oxide synthase that leads to inhibition of effector T cell and augmentation of Tregs [40]. A recent study in mice revealed that HSCs have an immunoprotective capacity towards transplanted hepatocytes. Four weeks after human hepatocytes were co-transplanted with human HSCs into mouse livers, human albumin positive hepatocytes were two to six times more numerous and had enhanced hepatocyte engraftment [41]. A study showed that co-transplantation of HSCs with allogenic pancreatic islets in humans protected islet grafts from rejection without the need for IS [40].

## 7. LSECs

Sinusoids are lined by LSECs. They are 5–7 µm in diameter and provide close contact with lymphocytes for effective immune interaction with LSECs. LSECs express minimal levels of CD80, CD86, CD40, and MHC class II hence are poor stimulators of T cells [42]. T cells trigger increased expression of co-inhibitory B7-H1 molecules exclusively on LSECs. These B7-H1 molecules interact with the PD-1 surface receptor on T cells and regulates T cell activation to induce tolerance [43]. LSECs secrete adhesion molecules that facilitate the sequestration of activated CD8 T cells. These alloantigen T cells are exposed to IL-10, PD-L1, and FasL1 that promote their destruction [27]. LSECs secrete LSECtin which interacts with CD44 on T cells and inhibits the secretion of IL-2 and IFNγ [44]. These functions of LSECs are shown to be preserved better in spontaneously accepted rat liver grafts [45].

## 8. KCs

KCs are resident macrophages in the liver described by Karl von Kupffer. They mature from circulating bone marrow-derived monocytes and account for 80–90% of tissue macrophages present in the body and directly interact with passenger leukocytes [46]. After liver transplantation, donor KCs migrate into the recipient’s lymph node and are replaced by recipient-derived monocytes [47]. A study by Sun et al. showed that recipient KCs recovered from chronically accepted hepatic allografts have increased FasL as well as IL-10 and TGF-β production. These products induce increased apoptosis of T cells compared to an acute rejection model [48,49]. In addition, prolonged graft survival is seen in rats with the infusion of KCs from chronically accepted liver grafts prior to transplantation [48]. Another study showed reduced TNFα from liver transplants pretreated with KCs, which prevented apoptosis of hepatocytes and enhanced the graft survival rate [50]. This further strengthens the immunomodulatory role of KCs on liver transplantation immune tolerance.

## 9. Hepatic NK Cells

NK cells, defined by the presence of CD56 and absence of CD3, are extensively present in the healthy liver [51]. Hepatic NK cells are continuously recruited from the peripheral blood into the liver where they can potentially differentiate into liver-specific NK cells [52]. While recipient NK cells have been shown to trigger organ rejection [53], Harmon et al. hypothesized that donor NK cells may prevent rejection [54]. High levels of NK cell-specific gene expression has been found in peripheral blood samples from transplant recipients with OT however they failed to identify whether the NK cells were of donor or recipient origin [55]. The exact role of donor NK cells in suppressing and regulating graft rejection has not been clearly defined. Activated alloantigen CD4 and CD8 T cells express NKG2DLs that make them susceptible to NK cell lysis in vitro and this could play a role in promoting tolerance [56].

## 10. Candidates for Cellular Therapy

### 10.1. Tregs

Tregs were first reported by Sakaguchi et al. in thymectomized mice [57]. They consist of 5–10% of CD4+ T cells and are broadly classified as thymic-derived and peripherally induced Tregs [58]. CD4, FoxP3, high levels of CD25, and low levels of CD127 are expressed on the surface. The CD25 marker serves as an “IL-2 sink” and consumes IL-2 which is the key cytokine required for the differentiation and maintenance of Tregs and stabilization of FoxP3 expression. The preferential consumption of IL-2 by Tregs suppresses the expansion of effector T cells [59]. Once activated, Tregs travel to the site of inflammation and prevent collateral tissue destruction by CD4+ and CD8+ T cells. Tregs secrete IL-10 [60], IL-35 [60], TGF β [61], and cAMP [62] to create an immunotolerant environment. Through the secretion of granzymes and perforins, they cause apoptosis of effector T cells [63,64]. CTLA4 is an essential negative regulator of T cell response on the surface of Tregs; the interaction between CTLA4 and costimulatory CD80/86 ligands on the surface of APCs activates indoleamine dioxygenase (IDO) in APCs. IDO depletes tryptophan in the local environment and consequently inhibits T effector cells [65] (Figure 1).

Tregs control transplant rejection by first migrating to the graft to induce graft tolerance and then drain into lymph nodes and help maintain tolerance [66]. Two types of Tregs can be expanded for clinical use: antigen-specific Tregs and polyclonal Tregs. On one hand, antigen-specific Tregs have specificity to one antigen and suppress effector T cells of distinct antigenic specificity making them more effective [67]. On the other hand, polyclonal Tregs are easier to manufacture [67]. Interestingly, Tregs demonstrate “infectious tolerance” [68], where the suppressive capacity of Tregs is transferred and can lead to the generation and expansion of a new population of antigen-specific Tregs distinct from the original Treg population. This occurs through the production of cytokines which induces the conversion of naïve T cells into Tregs [17]. Thus, tolerance persists much longer than the longevity of Tregs used.

The first clinical trial of T-cell-based therapy to induce OT was conducted by Todo et al., using an ex vivo-generated regulatory T-cell-enriched cell product in 10 adult patients post-LT [69]. The cultured Treg-enriched product significantly inhibited the proliferation of recipient lymphocytes in an ex-vivo Mixed Lymphocyte Reaction assay. In the human trial, the Treg product was administered on post-op day 13 along with a standard IS regimen including steroids, mycophenolate mofetil, and tacrolimus. IS drugs were gradually discontinued over 18 months. All 10 recipients maintained stable graft function. Seven patients with non-immunological liver diseases successfully achieved weaning and did not require ISs between 16 and 33 months. The other three recipients with autoimmune liver diseases developed mild rejection during weaning and were resumed on IS.

King’s group [23] proved the safety, applicability, and biological activity of autologous Treg adoptive transfer in humans in cadaveric liver transplantation. Circulating polyclonal Tregs were isolated from the recipient’s blood and expanded ex-vivo using IL-2 and rapamycin [70]. In three patients, Tregs were harvested pre-transplant, standard IS was administered for three months post-transplant, a biopsy was performed to exclude subclinical allograft rejection, and then Tregs were infused. In six patients, Tregs were harvested 8–14 months after transplant, with simultaneous liver biopsy, and the patients received expanded Treg infusion two months later. In recipients who received a single dose of 4.5 million Tregs/kg, a gradual decrease in T cell responses directed against donor cells was observed. Although they were not successful in weaning patients off IS, the development of donor-specific hyporesponsiveness is considered one of the hallmarks of transplantation tolerance [23].

Tregs demonstrate promise for targeted allograft tolerance and an alternative to classical IS. The first successful pilot study by Todo led to multiple OT induction trials currently using Treg cell therapy in LT [17]. Tregs do not persist in large numbers in the circulation after infusion due to low IL-2 availability. Even if they persist, a study revealed that Tregs may acquire a pro-inflammatory T effector phenotype as a consequence of FoxP3 instability [71]. Hence, their long-term viability needs to be addressed. In addition, due to the high frequency of alloreactive T effector cells post-transplant, Treg administration alone is insufficient and a combined regimen of lymphodepletion and donor-alloantigen-reactive Tregs is most effective in long-term graft survival [72]. Chimeric antigen receptor (CAR) T cells could be a promising method for the production of antigen-specific Tregs over polyclonal Tregs [73]. These engineered Tregs have a superior allospecific immune response and proliferation capacity [73]. Noyon et al. successfully transduced natural Tregs to donor-specific CAR Tregs without altering their regulatory phenotype. These CAR Tregs were far more potent in preventing rejection after a skin transplant in a mouse and could be applied to produce potent donor-antigen-specific Tregs after LT [73].

### 10.2. DCregs

DCs are stellate-shaped APCs expressing CD11c and CD1a. They arise from the bone marrow and seed into peripheral tissue. Mature DCs produce pro-inflammatory cytokines, while liver-derived DCs have a lower maturity status and produce high levels of regulatory factors including IL-10, IL-27, retinoic acid, and prostaglandin E2 [74]. DCs with immune tolerance properties are called DCregs; these cells express high levels of MHC, T cell co-inhibitory ligands (PDL-1), and death-inducing ligand (FasL) with low expression of co-stimulatory molecules [75]. DCregs are immature cells that induce apoptosis of alloantigen-specific T cells through the Fas/FasL pathway and IDO and generate Tregs and Bregs through the production of IL-10 and TGF β [76,77] (Figure 2).

In vitro generation of DCregs has been achieved by utilizing cytokines (IL-10, IL-4, TGF-β, and VEGF), pharmaceutical agents (aspirin, PGE2, histamine, β2 agonists, corticosteroids), and IS (cyclosporin A, rapamycin, and mycophenolate mofetil) [78]. The first and only clinical trial testing the efficacy of a single infusion of donor-derived DCregs in LT recipients is currently in phase I/II [79,80]. Monocytes from prospective liver donors were cryopreserved, then thawed and treated with L-glutamine, GM-CSF, IL-4, Vitamin D3, and IL-10. DCregs in this study exhibited a tolerogenic profile of high PDL1 and IL-10, resistance to maturation, and potential to modulate alloreactive T cell responses. Fifteen live donor LT donor-recipient pairs were recruited between 2017 and 2020 at the University of Pittsburgh and received a single infusion of these donor-derived DCregs one week prior to transplantation [79,80]. The timing of infusion of donor-derived DCs seven days prior to transplant has been supported by many prior studies; this timing is based on the hypothesis that DCs are most involved in the very early stages of the immune response and hyperacute rejection [38,81,82,83]. DCreg infusion in transplantation has resulted in decreased memory CD8+ T cells and increased Tregs in circulation [80]. This effective blunting of memory cells is a major victory in the promotion of long-term allograft survival [84]. Intact donor-derived DCregs with preserved regulatory phenotype were detected in the circulation shortly after their infusion, but there was no long-term persistence of donor DCregs. The Pittsburgh Study (NCT03164265) is yet to report on the post-transplant clinical and immunological outcomes, and whether this approach to donor DCreg therapy could result in the enhanced success of early IS withdrawal [80]. The estimated completion date of this study is June 2023.

Of note, ex-vivo expansion of DCregs yielded high numbers of viable cells, constituting a possible advantage over other types and sources of regulatory cells (e.g., autologous Tregs) which require several weeks to expand. Currently, donor-derived DCregs are restricted to live-donor organ transplantation and alternative therapies such as autologous DCregs loaded with donor antigen are promising for deceased donors [85]. Genetically engineered DCs may be a feasible approach to improve the therapy of allograft rejection as DCs can be virally transduced to express various immunosuppressive molecules, such as IL-10 and TGFβ, to further enhance their tolerogenic potential [86].

## 11. MSCs

Mesenchymal stem cells (MSCs) are nonhematopoietic multipotent progenitor cells that play a role in tissue remodeling and reside in various tissues. The International Society for Cell Therapy committee defines MSCs as (1) adhering to plastic and being fibroblast-like after culture in vitro; (2) being positive for CD105, CD73, and CD90 but negative for CD45, CD34, CD14, CD19, or HLA-DR by flow cytometry; (3) differentiating in vitro into osteoblasts, adipocytes, and chondroblasts [87]. MSCs inhibit the allogenic T cell response and IL-2 induced proliferation, cytotoxicity, and cytokine secretion in NK cells [88]. MSCs with high IDO activity promote IL-10-producing macrophages and inhibit the maturation of DCs [89,90]. Secretion of HLA-G5 promotes the generation and activation of Tregs [91] (Figure 3).

MSCs are not immunogenic as they do not express HLA hence they evade recipient T cells and can be a potent cellular therapy tool regardless of the immunogenic differences between MSCs and recipients [92]. In the transplant setting, the liver allograft environment is rich in pro-inflammatory cytokines, and IFNγ pretreated MSCs show upregulation of PDL1, MHC I, MHC II, and CD54 which enhances their immunoregulatory capacity [68]. Significant improvement in graft survival time and Treg upregulation was seen in rats after LT with early MSC treatment [93]. However, human trials have been less definitive. Detry et al. conducted the first clinical trial to assess the tolerability, feasibility, and safety of a single MSC infusion after LT [94]. LT recipients received non-HLA-matched third-party MSCs on postop day three and were compared to a control group receiving LTs only. There was no impact on Tregs, CD4+ cells count zero, one month, and three months post-transplant when compared to the control. From the MSC group recipients who had a normal biopsy at six months, progressive weaning of IS was attempted [94]. In one patient, tacrolimus and MMF withdrawal was performed without rejection and she remained off ISs for 12 months. In eight recipients, signs of graft rejection occurred during the withdrawal of ISs. Despite the suboptimal induction of OT, this study opens the way for further MSC-based trials in LT, perhaps with altered dosing or timing of MSC administration. Post-transplant MSC infusion was well tolerated, without evidence of pulmonary dysfunction or cytokine release syndrome and no cases of increased susceptibility to infection or de novo cancer occurred [95].

## 12. Future: Where do We Go from Here?

Induced pluripotent stem cells (iPSCs) are the result of reprogramming somatic cells to generate pluripotent progenitor cells and have great potential as a major cell source for producing various types of cells to induce tolerance in the field of transplantation [96]. iPSCs have an individualized approach to cell therapy since the patient’s own cells can be used to induce tolerance in transplants thus avoiding the need for IS. Senju et al. [97] succeeded in generating DCs from iPSCs and these cells were comparable to bone marrow-derived macrophages in terms of morphology and function [97]. This formed the foundation for which Zhang et al. [98] developed a feasible approach for iPS-DCregs which showed immune regulatory effects both in vitro and in vivo as well as the ability to generate Tregs in vitro [98]. MDSCs (myeloid-derived stem cells) exert immunosuppressive functions and have also been produced from iPSCs [99]. They inhibit T cell response and the secretion of IFNγ in a mixed lymphocytic reaction assay as well as in vivo in a mouse model [99]. These studies highlight the potential of iPS-DCregs and iPS-MDSCs as a key resource for cell therapy for inducing transplantation tolerance. Another potential therapeutic entity involves the use of extracellular vesicles (EVs) in the regulation of transplantation tolerance. EVs are nanosized, membrane-bound particles containing nucleic acid, lipids, or proteins that are released by many cell types. EVs containing transplant donor MHC can be received by recipient APCs, resulting in cross-dressing of APCs with allo-MHCs and immune stimulation [100]. However, new evidence suggests that EVs may also play a role in tolerance. A mouse model of MHC-mismatched liver transplantation developed tolerance because recipient DCs cross-dressed with donor EVs expressed high levels of co-inhibitory molecules PD-L1 and IL-10, leading to marked suppression of host alloreactivity [101]. Treg-EVs inhibited T-cell proliferation through the release of EVs transporting specific miRNAs and iNOS enzymes; when administered following rat kidney transplant, Treg-EVs prolonged the survival of the allograft [102]. Multiple preclinical and clinical trials are assessing EV therapeutic potential for a variety of pathological conditions, however, no clinical trials directly studying transplant tolerance are currently underway [103].

## 13. Conclusions

The induction of operational tolerance after transplant has been a critical goal ever since the first solid organ transplants were completed nearly 70 years ago. After the discovery of the liver as an immunologically privileged organ, our understanding of the immunomodulatory roles of the different cells in the liver continues to grow and we turn to cell therapy as a solution outside the purview of classic IS. We are closer than ever before to achieving the goal of operational tolerance in liver transplantation. Treg infusion after LT is currently undergoing multiple clinical trials. While we await these results, larger and longer-term trials are the next step in bringing this practice into widespread clinical use and hopefully will enhance the understanding of long-term Treg viability and the applications of chimeric antigen receptor technology. DCregs demonstrate a promising ability to blunt the memory immune response, as well as an important role in preventing hyperacute rejection; however, until clinical trial results are available in 2023, no definitive recommendation can be made. In addition, the role of recipient DCregs for deceased donor transplants needs further investigation. MSC administration has so far not demonstrated a consistent path for IS withdrawal, but more research into the timing and dosing of this powerful immune regulator could provide a promising cellular therapy for tolerance induction.

## Figures and Tables

**Figure 1 ijms-22-04016-f001:**
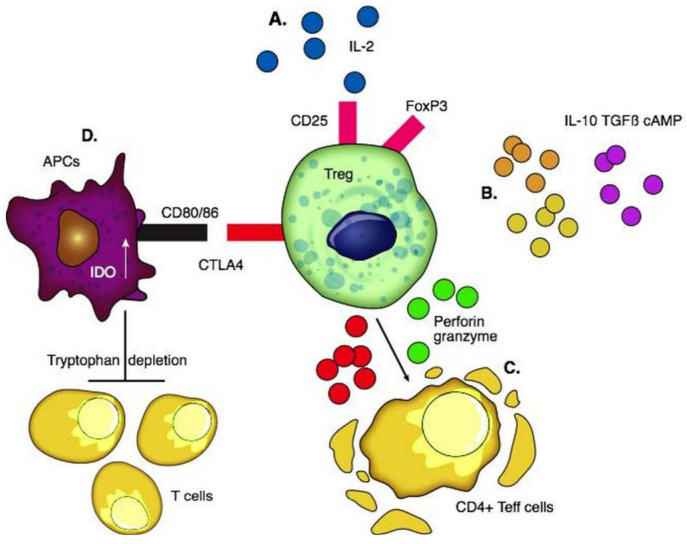
Functions of regulatory T cells. (**A**) Interaction of pro-inflammatory IL-2 with CD25 for the differentiation and maintenance of Tregs and stabilization of FoxP3. (**B**) Production of anti-inflammatory cytokines including IL-10, TGFβ, and cAMP to create an immunotolerant environment. (**C**) Secretion of granzymes and perforins causes apoptosis of CD4+T effector cells. (**D**) CTLA4 interacts with co-stimulatory CD80 and CD86 ligands on the surface of APCs and activates IDO in them which depletes tryptophan in the local environment and consequently inhibits the proliferation of T cells.

**Figure 2 ijms-22-04016-f002:**
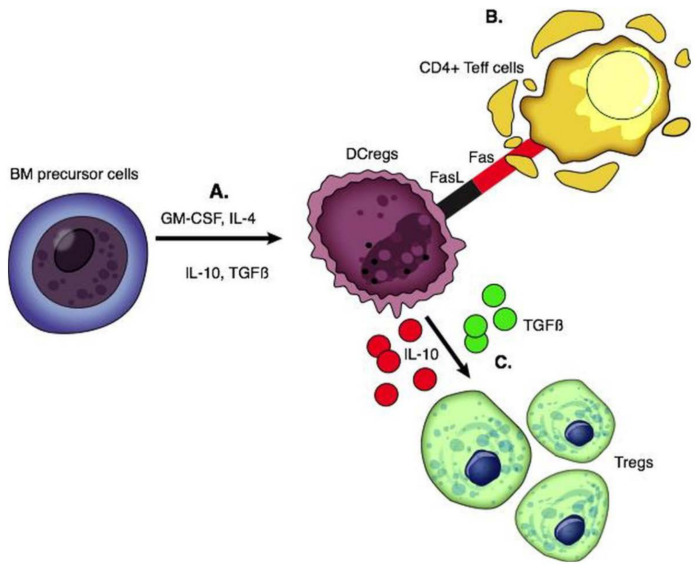
Functions of regulatory dendritic cells. (**A**) DCregs are generated from BM precursor cells by adding GM-CSF with IL-4, IL-10, and TGFβ. (**B**) DCregs express FasL and interact with the Fas on activated CD4 T effector cells which is associated with T cell apoptosis. (**C**) Production of IL-10 and TGFβ leads to the generation of Tregs.

**Figure 3 ijms-22-04016-f003:**
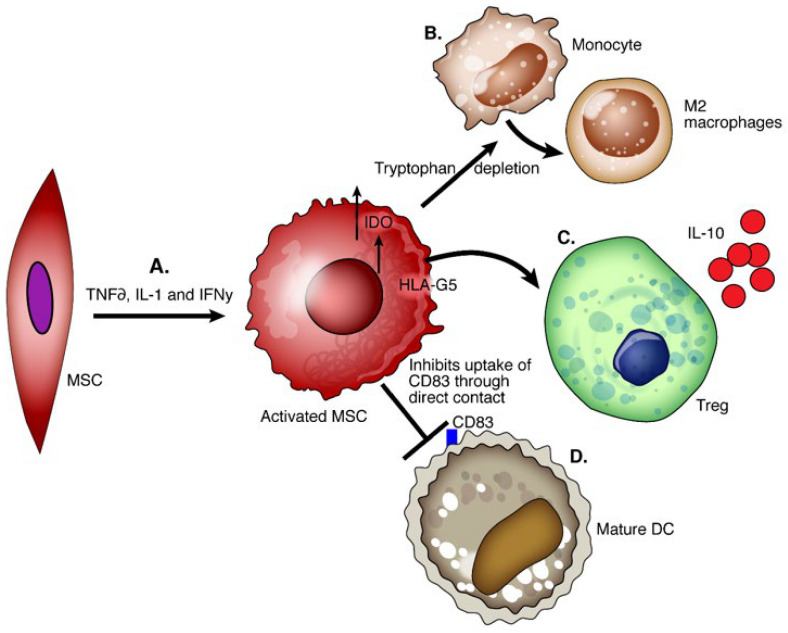
Functions of mesenchymal cells. (**A**) Mesenchymal cells pretreated with anti-inflammatory TNFα, IL-1, and IFNγ are activated and show enhanced immunoregulatory capacity. (**B**) The enzymatic activity of IDO and possibly other factors secreted by MSCs induces the conversion of monocytes into IL-10 producing M2 macrophages. (**C**) MSCs secrete HLA-G5 which generates Tregs and increases the production of IL-10. (**D**) MSCs through direct contact inhibit the uptake of CD83 necessary for the maturation of dendritic cells.

**Table 1 ijms-22-04016-t001:** Cell types of liver: distribution, essential functions, and markers.

Cell Type	% in Liver [29]	Primary Function	Location	Cell Surface Markers
Parenchymal cells (75–80%) [29]	Hepatocytes 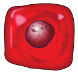	75–80%	Detoxification of NH3, metabolism of lipids, proteins, and carbohydrates, and synthesis of proteins	Hepatic lobules	Asioglycoprotein receptor 1, HepPar1, Arginase 1
Non-Parenchymal cells (5–6%) [29]	Kupffer cells 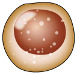	~33% of NPCs [29]	Macrophages with antigen-presenting function and phagocytosis	Within lumen of sinusoids	MHC II and co-stimulatory molecules
Mesenchymal cells 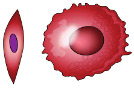	-	Tissue remodeling and repair	Within lumen of sinusoids	CD105, CD73, CD44, and CD90
Hepatic stellate cells 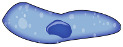	~3–5% of NPCs [30]	Maintain extracellular matrix, sinusoidal blood flow, and storage of vitamin A	Space of Disse	CD1d, CD86, and MHC II
Liver sinusoidal endothelial cells 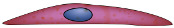	~44% of NPCs [29]	Endocytosis to remove pathogens and antigen-presenting capacity	Lining of sinusoids	CD80, CD86, CD40, and MCH I and II
Cholangiocytes 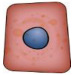	~3–5% of total liver cells [31]	Secretion of bile	Lining of intrahepatic and extrahepatic bile duct system	CK19, CK18, CK7, Sox9, OPN, and EpCAM
Intrahepatic lymphocytes 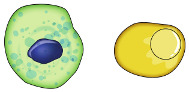	20% of NPCs [28]	Innate and adaptive immunity	Within lumen of sinusoids	B cells—CD19, CD20T cells—CD3, CD4NK cells—CD56NKT cells—CD56, CD3Tregs—CD4, CD25, FoxP3
Dendritic cells 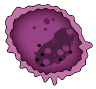	<1% of NPCs	Antigen presenting function and phagocytosis	Perisinusoidal	CD11c, CD1a

## Data Availability

Data sharing not applicable.

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
