# Peer review of "Cell-Mediated Therapies to Facilitate Operational Tolerance in Liver Transplantation"

_ijms, 2021, doi:10.3390/ijms22084016_

Round 1
Reviewer 1 Report
The manuscript by Ellias et al. reviews the latest cell based approaches to support operational tolerance after orthotopic liver transplantation.
The review focusses on the immunoprivileged status of the liver and provides an overview about the particular role of the liver cell types related to the tolergenicity. Additionally, the manuscript integrates literature from pre-clinical and clinical trails using cell therapeutics to foster the operational tolerance.
The manuscript is well written and the figures illustrate the functions of the cell therapeutics to underline the possible mechanisms.
line 376: 12. Future...: Please, may the authors consider to include the exosome topic or comment on that issue (for example: doi: 10.5500/wjt.v10.i11.330)
minor points:
line 55: Please check cit. format
line 77 and 78; 329: Please check the status of the clinical trails, it would be helpfull to the readers to know the Phase of the trails.
line 121: please check the number of lymphocytes
line 227; 249: please check the format of the cit.
line 257: please check period
line 266, 384: please check cit. format
Fig. 3: please check IDO and HLA-G5 hardly visible within the activated MSC. D. CD83 is missing, please check if CD83 could be add
Reviewer 2 Report
The manuscript "Cell mediated therapies to facilitate operational tolerance in liver transplantation" by Ellias et al. reviews the latest cell based approaches to support operational tolerance after orthotopic liver transplantation.
Authors have achieved a good revision from the current state of cell therapy to induce tolerance in liver transplant in the review "Cell mediated therapies to facilitate operational tolerance in liver transplantation". I accept the manuscript in the current form, however there are some format mistakes with the references which has to be properly reviewed by the authors. For example in line 55 or 227.
Author Response
Response to Reviewer 2 Comments
Point 1: there are some format mistakes with the references which has to be properly reviewed by the authors. For example in line 55 or 227.
Response 1: line 54: [] added to reference 10
Line 220-249 - Format of citations changed for reference [57], [60], [61], [62]
Line 266: Reference [70] format changed
Line 383: Reference [98] format changed